# Visual Dynamics: Probabilistic Future Frame Synthesis via Cross Convolutional Networks

**Tianfan Xue\***[1]    **Jiajun Wu\***[1]    **Katherine L. Bouman**[1]    **William T. Freeman**[1,2]
[1] Massachusetts Institute of Technology        [2] Google Research
{tfxue, jiajunwu, klbouman, billf}@mit.edu

## Abstract

We study the problem of synthesizing a number of likely future frames from a single input image. In contrast to traditional methods, which have tackled this problem in a deterministic or non-parametric way, we propose to model future frames in a probabilistic manner. Our probabilistic model makes it possible for us to sample and synthesize many possible future frames from a single input image. To synthesize realistic movement of objects, we propose a novel network structure, namely a *Cross Convolutional Network*; this network encodes image and motion information as feature maps and convolutional kernels, respectively. In experiments, our model performs well on synthetic data, such as 2D shapes and animated game sprites, as well as on real-world video frames. We also show that our model can be applied to visual analogy-making, and present an analysis of the learned network representations.

## 1   Introduction

From just a single snapshot, humans are often able to imagine how a scene will visually change over time. For instance, due to the pose of the girl in Figure 1, most would predict that her arms are stationary but her leg is moving. However, the exact motion is often unpredictable due to an intrinsic ambiguity. Is the girl's leg moving up or down? In this work, we study the problem of *visual dynamics*: modeling the conditional distribution of future frames given an observed image. We propose to tackle this problem using a probabilistic, content-aware motion prediction model that learns this distribution without using annotations. Sampling from this model allows us to visualize the many possible ways that an input image is likely to change over time.

Modeling the conditional distribution of future frames given only a single image as input is a very challenging task for a number of reasons. First, natural images come from a very high dimensional distribution that is difficult to model. Designing a generative model for realistic images is a very challenging problem. Second, in order to properly predict motion distributions, the model must first learn about image parts and the correlation of their respective motions in an unsupervised fashion.

In this work, we tackle the visual dynamics problem using a neural network structure, based on a variational autoencoder [Kingma and Welling, 2014] and our newly proposed cross convolutional layer. During training, the network observes a set of consecutive image pairs in videos, and automatically infers the relationship between them without any supervision. During testing, the network then predicts the conditional distribution, $P(J|I)$, of future RGB images $J$ (Figure 1b) given an RGB input image $I$ that was not in the training set (Figure 1a). Using this distribution, the network is able to synthesize multiple different image samples corresponding to possible future frames of the input image (Figure 1c). Our network contains a number of key components that contribute to its success:

- We use a conditional variational autoencoder to model the complex conditional distribution of future frames [Kingma and Welling, 2014, Yan et al., 2016]. This allows us to approximate a sample, $J$, from the distribution of future images by using a trainable function $J = f(I, z)$.

---

∗ indicates equal contributions.

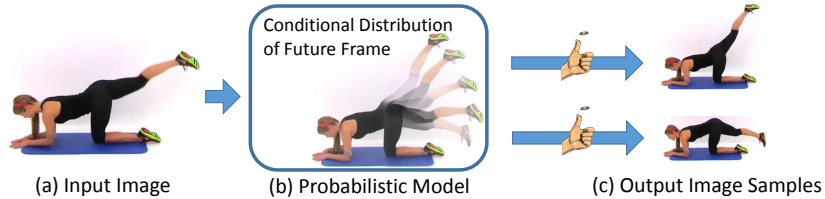

(a) Input Image       (b) Probabilistic Model       (c) Output Image Samples

Figure 1: Predicting the movement of an object from a single snapshot is often ambiguous. For instance, is the girl's leg in (a) moving up or down? We propose a probabilistic, content-aware motion prediction model (b) that learns the conditional distribution of future frames. Using this model we are able to synthesize various future frames (c) that are all consistent with the observed input (a).

> The argument $z$ is a sample from a simple distribution, *e.g.* Gaussian, which introduces randomness into the sampling of $J$. This formulation makes the problem of learning the distribution much more tractable than explicitly modeling the distribution.

- We model motion using a set of image-dependent convolution kernels operating over an image pyramid. Unlike normal convolutional layers, these kernels vary between images, as different images may have different motions. Our proposed cross convolutional layer convolves image-dependent kernels with feature maps from an observed frame, to synthesize a probable future frame.

We test the proposed model on two synthetic datasets as well as a dataset generated from real videos. We show that, given an RGB input image, the algorithm can successfully model a distribution of possible future frames, and generate different samples that cover a variety of realistic motions. In addition, we demonstrate that our model can be easily applied to tasks such as visual analogy-making, and present an analysis of the learned network representations.

## 2 Related Work

**Motion priors** Research studying the human visual system and motion priors provides evidence for low-level statistics of object motion. Pioneering work by Weiss and Adelson [1998] found that the human visual system prefers slow and smooth motion fields. More recent work by Roth and Black [2005] analyzed the response of spatial filters applied to optical flow fields. Fleet et al. [2000] also found that a local motion field can be represented by a linear combination of a small number of bases. All these works focus on the distribution of a motion field itself without considering any image information. On the contrary, our context-aware model captures the relationship between an observed image and its motion field.

**Motion or future prediction** Our problem is closely related to the motion or feature prediction problem. Given an observed image or a short video sequence, models have been proposed to predict a future motion field [Liu et al., 2011, Pintea et al., 2014, Xue et al., 2014, Walker et al., 2015, 2016], a future trajectory of objects [Walker et al., 2014, Wu et al., 2015], or a future visual representation [Vondrick et al., 2016b]. Most of these works use deterministic prediction models [Pintea et al., 2014, Vondrick et al., 2016b]. Recently, and concurrently with our own work, Walker et al. [2016] found that there is an intrinsic ambiguity in deterministic prediction, and propose a probabilistic prediction framework. Our model is also a probabilistic prediction model, but it directly predicts the pixel values, rather than motion fields or image features.

**Parametric image synthesis** Early work in parametric image synthesis mostly focus on texture synthesis using hand-crafted features [Portilla and Simoncelli, 2000]. More recently, works in image synthesis have begun to produce impressive results by training variants of neural network structures to produce novel images [Gregor et al., 2015, Xie et al., 2016a,b, Zhou et al., 2016]. Generative adversarial networks [Goodfellow et al., 2014, Denton et al., 2015, Radford et al., 2016] and variational autoencoders [Kingma and Welling, 2014, Yan et al., 2016] have been used to model and sample from natural image distributions. Our proposed algorithm is also based on the variational autoencoder, but unlike in this previous work, we also model temporal consistency.

**Video synthesis** Techniques that exploit the periodic structure of motion in videos have also been successful at generating novel frames from an input sequence. Early work in video textures proposed to shuffle frames from an existing video to generate a temporally consistent, looping image sequence [Schödl et al., 2000]. These ideas were later extended to generate cinemagraphies [Joshi et al., 2012], seamlessly looping videos containing a variety of objects with different motion patterns [Agarwala et al., 2005, Liao et al., 2013], or video inpainting [Wexler et al., 2004]. While

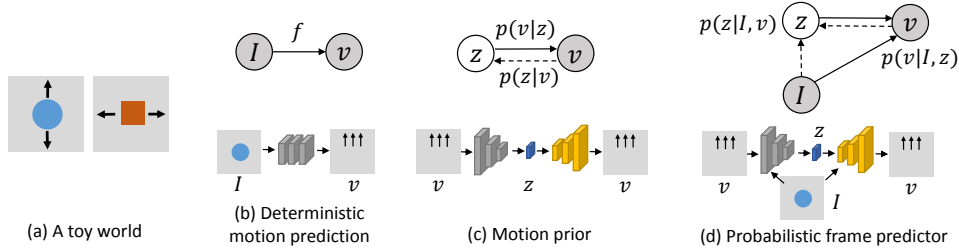

(a) A toy world     (b) Deterministic motion prediction     (c) Motion prior     (d) Probabilistic frame predictor

Figure 2: A toy world example. See Section 3.2 for details.

high-resolution and realistic looking videos are generated using these techniques, they are often limited to periodic motion and require an input reference video. In contrast, we build an image generation model that does not require a reference video at test time.

Recently, several network structures have been proposed to synthesize a new frame from observed frames. They infer the future motion either from multiple previous frames Srivastava et al. [2015], Mathieu et al. [2016], user-supplied action labels Oh et al. [2015], Finn et al. [2016], or a random vector Vondrick et al. [2016a]. In contrast to these approaches, our network takes a single frame as input and learns the distribution of future frames without any supervision.

## 3 Formulation

### 3.1 Problem Definition

In this section, we describe how to sample future frames from a current observation image. Here we focus on next frame synthesis; given an RGB image $I$ observed at time $t$, our goal is to model the conditional distribution of possible frames observed at time $t + 1$.

Formally, let $\{(I^{(1)}, J^{(1)}), \ldots, (I^{(n)}, J^{(n)})\}$ be the set of image pairs in the training set, where $I^{(i)}$ and $J^{(i)}$ are images observed at two consecutive time steps. Using this data, our task is to model the distribution $p_\theta(J|I)$ of all possible next frames $J$ for a new, previously unseen test image $I$, and then to sample new images from this distribution. In practice, we choose not to directly predict the next frame, but instead to predict the difference image $v = J - I$, also known as the Eulerian motion, between the observed frame $I$ and the future frame $J$; these two problems are equivalent. The task is then to learn the conditional distribution $p_\theta(v|I)$ from a set of training pairs $\{(I^{(1)}, v^{(1)}), \ldots, (I^{(n)}, v^{(n)})\}$.

### 3.2 A Toy Example

Consider a simple toy world that only consists of circles and squares. All circles move vertically, while all squares move horizontally, as shown in the Figure 2(a). Although in practice we choose $v$ to be the difference image between consecutive frames, for this toy example we show $v$ as a 2D motion field for a more intuitive visualization. Consider the three models shown in Figure 2.

**(1) Deterministic motion prediction** In this structure, the model tries to find a deterministic relationship between the input image and object motion (Figure 2(b)). To do this, it attempts to find a function $f$ that minimizes the reconstruction error $\sum_i ||v^{(i)} - f(I^{(i)})||$ on a training set. Thus, it cannot capture the multiple possible motions that a shape can have, and the algorithm can only learn a mean motion for each object. In the case of zero-mean, symmetric motion distributions, the algorithm would produce an output frame with almost no motion.

**(2) Motion prior** A simple way to model the multiple possible motions of future frames is to use a variational autoencoder [Kingma and Welling, 2014], as shown in Figure 2(c). The network consists of an encoder network (gray) and a decoder network (yellow), and the latent representation $z$ encodes the intrinsic dimensionality of the motion fields. A shortcoming of this model is that it does not see the input image during inference. Therefore, it will only learn a global motion field of both circles and squares, without distinguishing the particular motion pattern for each class of objects.

**(3) Probabilistic frame predictor** In this work, we combine the deterministic motion prediction structure with a motion prior, to model the uncertainty in a motion field and the correlation between motion and image content. We extend the decoder in (2) to take two inputs, the intrinsic motion representation $z$ and an image $I$ (see the yellow network in Figure 2(d), which corresponds to $p(v|I, z)$). Therefore, instead of modeling a joint distribution of motion $v$, it will learn a conditional distribution of motion given the input image $I$.

In this toy example, since squares and circles only move in one (although different) direction, we would only need a scalar $z \in \mathbb{R}$ for encoding the velocity of the object. The model is then able to infer the location and direction of motion conditioned on the shape that appears in the input image.

### 3.3 Conditional Variational Autoencoder

In this section, we will formally derive the training objective of our model, following the similar derivations as those in Kingma and Welling [2014], Kingma et al. [2014], Yan et al. [2016]. Consider the following generative process that samples a future frame conditioned on an observed image, $I$. First, the algorithm samples the hidden variable $z$ from a prior distribution $p_z(z)$; in this work, we assume $p_z(z)$ is a multivariate Gaussian distribution where each dimension is i.i.d. with zero-mean and unit-variance. Then, given a value of $z$, the algorithm samples the intensity difference image $v$ from the conditional distribution $p_\theta(v|I, z)$. The final image, $J = I + v$, is then returned as output.

In the training stage, the algorithm attempts to maximize the log-likelihood of the conditional marginal distribution $\sum_i \log p(v^{(i)}|I^{(i)})$. Assuming $I$ and $z$ are independent, the marginal distribution is expanded as $\sum_i \log \int_z p(v^{(i)}|I^{(i)}, z)p_z(z)dz$. Directly maximizing this marginal distribution is hard, thus we instead maximize its variational upper-bound, as proposed by Kingma and Welling [2014]. Each term in the marginal distribution is upper-bounded by

$$\mathcal{L}(\theta, \phi, v^{(i)}|I^{(i)}) \approx -D_{\text{KL}}(q_\phi(z|v^{(i)}, I^{(i)})||p_z(z)) + \frac{1}{L}\sum_{l=1}^{L}\left[\log p_\theta(v^{(i)}|z^{(i,l)}, I^{(i)})\right], \quad (1)$$

where $D_{\text{KL}}$ is the KL-divergence, $q_\phi(z|v^{(i)}, I^{(i)})$ is the variational distribution that approximates the posterior $p(z|v^{(i)}, I^{(i)})$, and $z^{(i,l)}$ are samples from the variational distribution. For simplicity, we refer to the conditional data distribution, $p_\theta(\cdot)$, as the *generative model*, and the variational distribution, $q_\phi(\cdot)$, as the *recognition model*.

We assume Gaussian distributions for both the generative model and recognition model[*], where the mean and variance of the distributions are functions specified by neural networks, that is[†]:

$$p_\theta(v^{(i)}|z^{(i,l)}, I^{(i)}) = \mathcal{N}(v^{(i)}; f_{\text{mean}}(z^{(i,l)}, I^{(i)}), \sigma^2\mathbf{I}), \quad (2)$$

$$q_\phi(z^{(i,l)}|v^{(i)}, I^{(i)}) = \mathcal{N}(z^{(i,l)}; g_{\text{mean}}(v^{(i)}, I^{(i)}), g_{\text{var}}(v^{(i)}, I^{(i)})), \quad (3)$$

where $\mathcal{N}(\,\cdot\,; a, b)$ is a Gaussian distrubtion with mean $a$ and variance $b$. $f_{\text{mean}}$ is a function that predicts the mean of the generative model, defined by the generative network (the yellow network in Figure 2(d)). $g_{\text{mean}}$ and $g_{\text{var}}$ are functions that predict the mean and variance of the recognition model, respectively, defined by the recognition network (the gray network in Figure 2(d)). Here we assume that all dimensions of the generative model have the same variance $\sigma^2$, where $\sigma$ is a hand-tuned hyper parameter. In the next section, we will describe the details of both network structures.

## 4 Method

In this section we present a trainable neural network structure, which defines the generative function $f_{\text{mean}}$ and recognition functions $g_{\text{mean}}$, and $g_{\text{var}}$. Once trained, these functions can be used in conjunction with an input image to sample future frames. We first describe our newly proposed cross convolutional layer, which naturally characterizes a layered motion representation [Wang and Adelson, 1993]. We then explain our network structure and demonstrate how we integrate the cross convolutional layer into the network for future frame synthesis.

### 4.1 Layered Motion Representations and Cross Convolutional Networks

Motion can often be decomposed in a layer-wise manner [Wang and Adelson, 1993]. Intuitively, different semantic segments in an image should have different distributions over all possible motions; for example, a building is often static, but a river flows.

To model layered motion, we propose a novel cross convolutional network (Figure 3). The network first decomposes an input image pyramid into multiple feature maps through an image encoder (Figure 3(c)). It then convolves these maps with different kernels (Figure 3(d)), and uses the outputs to synthesize a difference image (Figure 3(e)). This network structure naturally fits a layered motion representation, as each feature map characterizes an image *layer* (note this is different from a network

---

[*]A complicated distribution can be approximated by a function of a simple distribution, e.g. Gaussian, which is referred as the reparameterization trick in [Kingma and Welling, 2014].

[†]Here the bold $\mathbf{I}$ denotes an identity matrix, whereas the normal-font $I$ denotes the observed image.

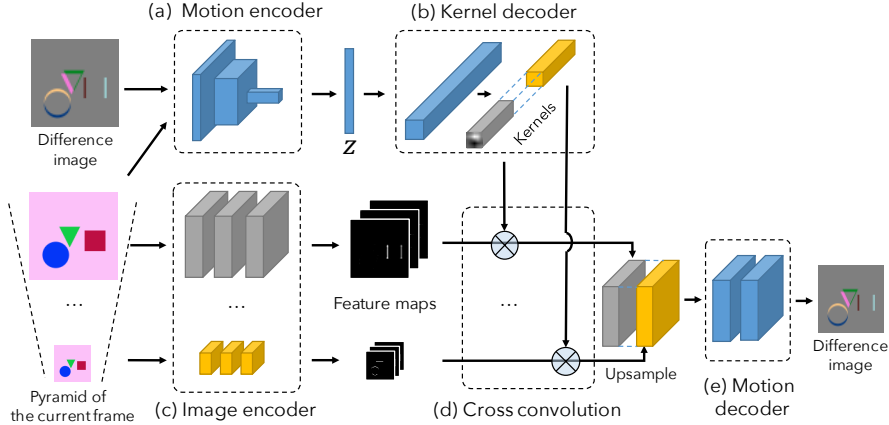

Figure 3: Our network consists of five components: (a) a motion encoder, (b) a kernel decoder, (c) an image encoder, (d) a cross convolution layer, and (e) a motion decoder. Our image encoder takes images at four scales as input. For simplicity, we only show two scales in this figure.

*layer*) and the corresponding kernel characterizes the motion of that layer. In other words, we model motions as convolutional kernels, which are applied to feature maps of images at multiple scales.

Unlike a traditional convolutional network, these kernels should not be identical for all inputs, as different images typically have different motions (kernels). We therefore propose a cross convolutional layer to tackle this problem. The cross convolutional layer does not learn the weights of the kernels itself. Instead, it takes both kernel weights and feature maps as input and performs convolution during a forward pass; for back propagation, it computes the gradients of both convolutional kernels and feature maps. Concurrent works from Finn et al. [2016], Brabandere et al. [2016] also explored similar ideas. While they applied the learned kernels on input images, we jointly learn feature maps and kernels without direct supervision.

### 4.2 Network Structure

As shown in Figure 3, our network consists of five components: (a) a motion encoder, (b) a kernel decoder, (c) an image encoder, (d) a cross convolutional layer, and (e) a motion decoder. The recognition functions $g_{\text{mean}}$ and $g_{\text{var}}$ are defined by the motion encoder, whereas the generative function $f_{\text{mean}}$ is defined by the remaining network.

During training, our variational motion encoder (Figure 3(a)) takes two adjacent frames in time as input, both at a resolution of $128 \times 128$, and outputs a 3,200-dimensional mean vector and a 3,200-dimensional variance vector. The network samples the latent motion representation $z$ using these mean and variance vectors. Next, the kernel decoder (Figure 3(b)) sends the $3{,}200 = 128 \times 5 \times 5$ tensor into two additional convolutional layers, producing four sets of 32 motion kernels of size $5 \times 5$. Our image encoder (Figure 3(c)) operates on four different scaled versions of the input image $I$ ($256 \times 256$, $128 \times 128$, $64 \times 64$, and $32 \times 32$). The output sizes of the feature maps in these four channels are $32 \times 64 \times 64$, $32 \times 32 \times 32$, $32 \times 16 \times 16$, and $32 \times 8 \times 8$, respectively. This multi-scale convolutional network allows us to model both global and local structures in the image, which may have different motions. See appendix for more details.

The core of our network is a cross convolutional layer (Figure 3(d)) which, as discussed in Section 4.1, applies the kernels learned by the kernel decoder to the feature maps learned by the image encoder, respectively. The output size of the cross convolutional layer is identical to that of the image encoder. Finally, our motion decoder (Figure 3(e)) uses the output of the cross convolutional layer to regress the output difference image.

**Training and testing details** During training, the image encoder takes a single frame $I^{(i)}$ as input, and the motion encoder takes both $I^{(i)}$ and the difference image $v^{(i)} = J^{(i)} - I^{(i)}$ as input, where $J^{(i)}$ is the next frame. The network aims to regress the difference image that minimizes the $\ell^2$ loss.

During testing, the image encoder still sees a single image $I$; however, instead of using a motion encoder, we directly sample motion vectors $z^{(j)}$ from the prior distribution $p_z(z)$. In practice, we use an empirical distribution of $z$ over all training samples as an approximation to the prior, as we find it produces better synthesis results. The network synthesizes possible difference images $v^{(j)}$ by taking

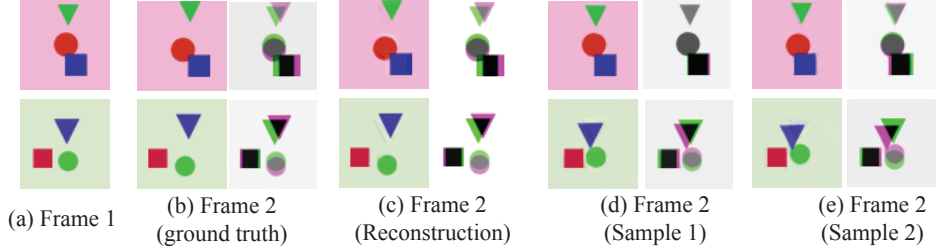

| (a) Frame 1 | (b) Frame 2 (ground truth) | (c) Frame 2 (Reconstruction) | (d) Frame 2 (Sample 1) | (e) Frame 2 (Sample 2) |

Figure 4: Results on the shapes dataset containing circles (C) squares (S) and triangles (T). For each 'Frame 2' we show the RGB image along with an overlay of green and magenta versions of the 2 consecutive frames, to help illustrate motion. See text and our project page for more details and a better visualization.

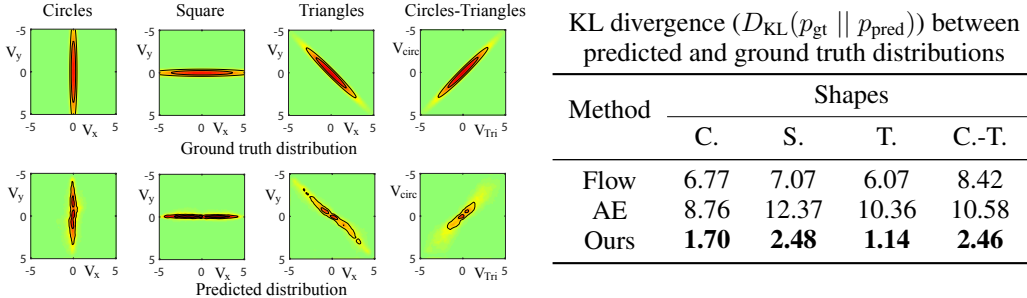

KL divergence ($D_{KL}(p_{gt} \parallel p_{pred})$) between predicted and ground truth distributions

| Method | Shapes | | | |
|---|---|---|---|---|
| | C. | S. | T. | C.-T. |
| Flow | 6.77 | 7.07 | 6.07 | 8.42 |
| AE | 8.76 | 12.37 | 10.36 | 10.58 |
| Ours | **1.70** | **2.48** | **1.14** | **2.46** |

Figure 5: Left: for each object, comparison between its ground-truth motion distribution and the distribution predicted by our method. Right: KL divergence between ground-truth distributions and distributions predicted by three different algorithms.

the sampled latent representation $z^{(j)}$ and an RGB image $I$ as input. We then generate a set of future frames $\{J^{(j)}\}$ from these difference images: $J^{(j)} = I + v^{(j)}$.

## 5  Evaluations

We now present a series of experiments to evaluate our method. All experimental results, along with additional visualizations, are also available on our project page[‡].

**Movement of 2D shapes**    We first evaluate our method using a dataset of synthetic 2D shapes. This dataset serves to benchmark our model on objects with simple, yet nontrivial, motion distributions. It contains three types of objects: circles, squares, and triangles. Circles always move vertically, squares horizontally, and triangles diagonally. The motion of circles and squares are independent, while the motion of circles and triangles are correlated. The shapes can be heavily occluded, and their sizes, positions, and colors are chosen randomly. There are 20,000 pairs for training, and 500 for testing.

Results are shown in Figure 4. Figure 4(a) and (b) show a sample of consecutive frames in the dataset, and Figure 4(c) shows the reconstruction of the second frame after encoding and decoding with the ground truth images. Figure 4(d) and (e) show samples of the second frame; in these results the network only takes the first image as input, and the compact motion representation, $z$, is randomly sampled. Note that the network is able to capture the distinctive motion pattern for each shape, including the strong correlation of triangle and circle motion.

To quantitatively evaluate our algorithm, we compare the displacement distributions of circles, squares, and triangles in the sampled images with their ground truth distributions. We sampled 50,000 images and used the optical flow package by Liu [2009] to calculate the movement of each object. We compare our algorithm with a simple baseline that copies the optical flow field from the training set ('Flow' in Figure 5 right); for each test image, we find its 10-nearest neighbors in the training set, and randomly transfer one of the corresponding optical flow fields. To illustrate the advantage of using a variational autoencoder over a standard autoencoder, we also modify our network by removing the KL-divergence loss and sampling layer ('AE' in Figure 5 right). Figure 5 shows our predicted distribution is very close to the ground-truth distribution. It also shows that a variational autoencoder helps to capture the true distribution of future frames.

---

[‡]Our project page: http://visualdynamics.csail.mit.edu

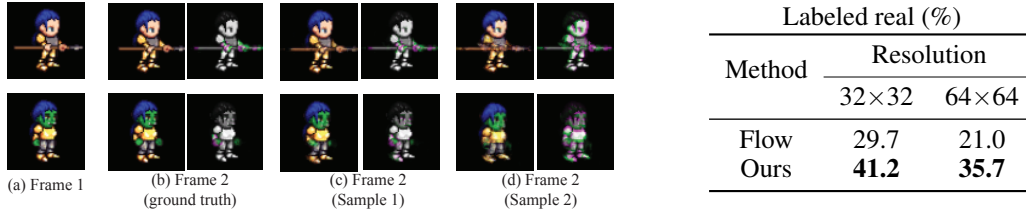

| (a) Frame 1 | (b) Frame 2 (ground truth) | (c) Frame 2 (Sample 1) | (d) Frame 2 (Sample 2) |

| | Labeled real (%) | |
| Method | Resolution | |
| | 32×32 | 64×64 |
| Flow | 29.7 | 21.0 |
| **Ours** | **41.2** | **35.7** |

Figure 6: Left: Sampling results on the Sprites dataset. Motion is illustrated using the overlay described in Figure 4. Right: Probability that a synthesized result is labeled as real by humans in Mechanical Turk behavioral experiments

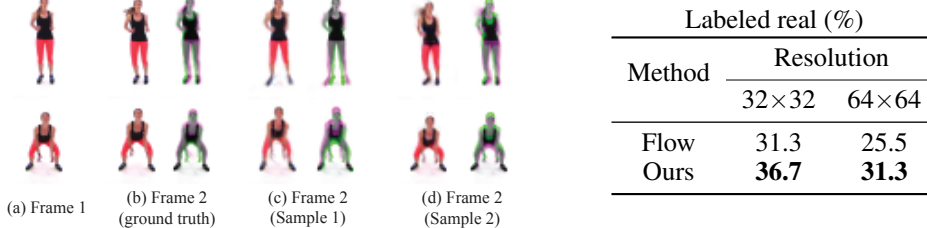

| (a) Frame 1 | (b) Frame 2 (ground truth) | (c) Frame 2 (Sample 1) | (d) Frame 2 (Sample 2) |

| | Labeled real (%) | |
| Method | Resolution | |
| | 32×32 | 64×64 |
| Flow | 31.3 | 25.5 |
| **Ours** | **36.7** | **31.3** |

Figure 7: Results on Exercise dataset. Left: Sampling results on *Exercise* dataset. Motion is illustrated using the overlay described in Figure 4. Right: probability that a synthesized result is labeled as real by humans in Mechanical Turk behavior experiments

**Movement of video game sprites**  We evaluate our framework on a video game sprites dataset[§], also used by Reed et al. [2015]. The dataset consists of 672 unique characters, and for each character there are 5 animations (spellcast, thrust, walk, slash, shoot) from 4 different viewpoints. Each animation ranges from 6 to 13 frames. We collect 102,364 pairs of neighboring frames for training, and 3,140 pairs for testing. The same character does not appear in both the training and test sets. Synthesized sample frames are shown in Figure 6. The results show that from a single input frame, our method can capture various possible motions that are consistent with those in the training set.

For a quantitative evaluation, we conduct behavioral experiments on Amazon Mechanical Turk. We randomly select 200 images, sample possible next frames using our algorithm, and show them to multiple human subjects as an animation side by side with the ground truth animation. We then ask the subject to choose which animation is real (not synthesized). An ideal algorithm should achieve a success rate of 50%. In our experiments, we present the animation in both the original resolution ($64 \times 64$) and a lower resolution ($32 \times 32$). We only evaluate on subjects that have a past approval rating of $> 95\%$ and also pass our qualification tests. Figure 6 shows that our algorithm significantly out-performs a baseline algorithm that warps an input image by transferring a randomly selected flow field from the training set. Subjects are more easily fooled by the $32 \times 32$ pixel images, as it is harder to hallucinate realistic details in high-resolution images.

**Movement in real videos captured in the wild**  To demonstrate that our algorithm can also handle real videos, we collect 20 workout videos from YouTube, each about 30 to 60 minutes long. We first apply motion stabilization to the training data as a pre-processing step to remove camera motion. We then extract 56,838 pairs of frames for training and 6,243 pairs for testing. The training and testing pairs come from different video sequences. Figure 7 shows that our framework works well in predicting the movement of the legs and torso. Additionally, Mechanical Turk behavioral experiments show that the synthesized frames are visually realistic.

**Zero-shot visual analogy-making**  Recently, Reed et al. [2015] studied the problem of inferring the relationship between a pair of reference images and synthesizing a new analogy-image by applying the inferred relationship to a test image. Our network is also able to preform this task, without even requiring supervision. Specifically, we extract the motion vector, $z$, from two reference frames using our motion encoder (Figure 3(a)). We then use the extracted motion vector $z$ to synthesize an analogy-image given a new test image.

Our network successfully transfers the motion in reference pairs to a test image. For example, in Figure 8(a), it learns that the character leans toward to the right, and in Figure 8(b) it learns that the girl spreads her feet apart. A quantitative evaluation is also shown in Figure 9. Even without

---

[§]Liberated pixel cup: http://lpc.opengameart.org

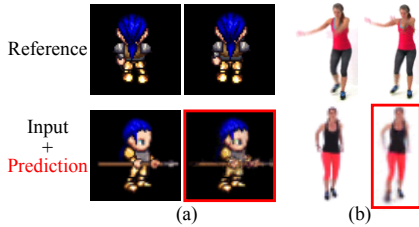

Reference

Input
+
Prediction

(a)　　　　　(b)

Figure 8: Visual analogy-making (predicted frames are marked in red)

| Model | spellcast | thrust | walk | slash | shoot | average |
|---|---|---|---|---|---|---|
| Add | 41.0 | 53.8 | 55.7 | 52.1 | 77.6 | 56.0 |
| Dis | 40.8 | 55.8 | 52.6 | 53.5 | 79.8 | 56.5 |
| Dis+Cls | 13.3 | 24.6 | 17.2 | **18.9** | 40.8 | 23.0 |
| Ours | **9.5** | **11.5** | **11.1** | 28.2 | **19.0** | **15.9** |

Figure 9: Mean squared pixel error on test analogies, by animation. The first three models (Add, Dis, and Dis+Cls) are from Reed et al. [2015].

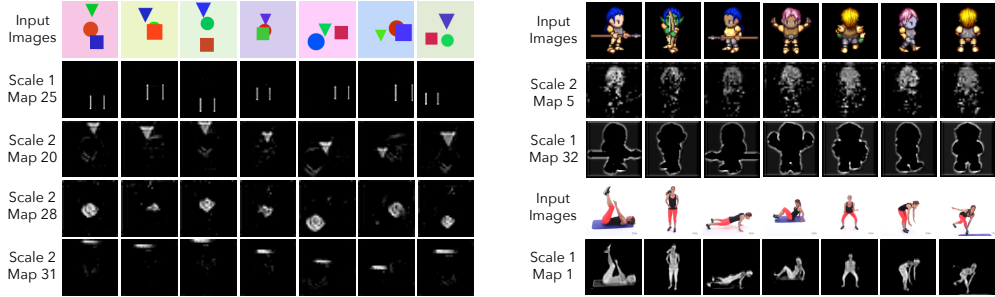

Figure 10: Learned feature maps on the shapes dataset (left), the sprites dataset (top right), and the exercise dataset (bottom right)

supervision, our method out-performs the algorithm by Reed et al. [2015], which requires visual analogy labels during training.

**Visualizing feature maps** We visualize the learned feature maps (see Figure 3(b)) in Figure 10. Even without supervision, our network learns to detect objects or contours in the image. For example, we see that the network automatically learns object detectors and edge detectors on the shape dataset. It also learns a hair detector and a body detector on the sprites and exercise datasets, respectively.

**Visualizing latent representations** By visualizing the latent representations of $z$ we have found that each dimension corresponds to a certain type of motion. For instance, in the excerise dataset, varying one dimension of $z$ causes the girl to stand-up and another causes her to move a leg. Please refer to our project page for this visualization.

**Dimension of latent representation** Although our latent motion representation, $z$, has 3,200 dimensions, its intrinsic dimensionality is much smaller. First, $z_{\text{mean}}$ is very sparse. The non-zero elements of $z_{\text{mean}}$ for each dataset are 299 in shapes, 54 in sprites, and 978 in exercise. Second, the independent components of $z$ are even fewer. We run principle component analysis (PCA) on the $z_{\text{mean}}$s obtained from a set of training images, and find that for each dataset, a small fraction of components cover at least 95% of the variance in $z_{\text{mean}}$ (5 in shapes, 2 in sprites, and 27 in exercise). This indicates that our network has learned a compact representation of motion in an unsupervised fashion, and encodes high-level knowledge using a small number of bits, rather than simply memorizing training samples. The KL-divergence criterion in Eq. 1 forces the latent representation, $z$, to carry minimal information, as discussed by Hinton and Van Camp [1993] and concurrently by Higgins et al. [2016].

# 6 Conclusion

In this paper, we have proposed a novel framework that samples future frames from a single input image. Our method incorporates a variational autoencoder for learning compact motion representations, and a novel cross convolutional layer for regressing Eulerian motion maps. We have demonstrated that our framework works well on both synthetic, and real-life videos.

More generally, results suggest that our probabilistic visual dynamics model may be useful for additional applications, such as inferring objects' higher-order relationships by examining correlations in their motion distributions. Furthermore, this learned representation could be potentially used as a sophisticated motion prior in other computer vision and computational photography applications.

**Acknowledgement** The authors thank Yining Wang for helpful discussions. This work is supported by NSF Robust Intelligence 1212849, NSF Big Data 1447476, ONR MURI 6923196, Adobe, and Shell Research. The authors would also like to thank Nvidia for GPU donations.

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
