[Supplementary Material]

## A.1 Derivation of Conditional Variation Autoencoder

In this section, we will formally derive how we obtain the training objective function in Eq. 1, following similar derivations in Kingma and Welling [2014], Kingma et al. [2014], Yan et al. [2016]. As mentioned in Section 3.3, the generative process that samples a difference image $v$ from a $\theta$-parametrized model, conditioned on an observed image $I$, consists of two steps. First, the algorithm samples the hidden variable $z$ from a prior distribution $p_z(z)$. Then, given a value of $z$, the algorithm samples the intensity difference image $v$ from the conditional distribution $p_\theta(v|I, z)$. This process is also described in the graphical model in Figure 2(d).

Given a set of training pairs $\{I^{(i)}, v^{(i)}\}$, the algorithm maximizes the log-likelihood of the conditional marginal distribution during training

$$\sum_i \log p(v^{(i)}|I^{(i)}). \tag{4}$$

Recall that $I$ and $z$ are independent as shown in the graphical model in Figure 2. Therefore, based on the Bayes' theorem, we have

$$p(v^{(i)}|I^{(i)}) = \frac{p_z(z)p_\theta(v^{(i)}|I^{(i)}, z)}{p(z|v^{(i)}, I^{(i)})}. \tag{5}$$

It is hard to directly maximizing the marginal distribution (Eq. 4). We therefore maximize its variational upper-bound instead, as proposed by Kingma and Welling [2014]. Let $q_\phi(z|v^{(i)}, I^{(i)})$ be the variational distribution that approximates the posterior $p(z|v^{(i)}, I^{(i)})$. Then each term in the marginal distribution is upper bounded as

$$\log p(v^{(i)}|I^{(i)}) = \mathbb{E}_{q_\phi(z|v^{(i)}, I^{(i)})}\left[\log p(v^{(i)}|I^{(i)})\right] \tag{6}$$

$$= \mathbb{E}_{q_\phi(z|v^{(i)}, I^{(i)})}\left[\log \frac{p_z(z)p_\theta(v^{(i)}|I^{(i)}, z)}{p(z|v^{(i)}, I^{(i)})}\right] \tag{7}$$

$$= \mathbb{E}_{q_\phi(z|v^{(i)}, I^{(i)})}\left[\log \frac{p_z(z)}{q_\phi(z|v^{(i)}, I^{(i)})}\right] + \mathbb{E}_{q_\phi(z|v^{(i)}, I^{(i)})}\left[\log \frac{q_\phi(z|v^{(i)}, I^{(i)})}{p(z|v^{(i)}, I^{(i)})}\right]$$
$$+ \mathbb{E}_{q_\phi(z|v^{(i)}, I^{(i)})}[\log p_\theta(v^{(i)}|I^{(i)}, z)] \tag{8}$$

$$= -D_{\mathrm{KL}}(q_\phi(z|v^{(i)}, I^{(i)})||p_z(z)) + D_{\mathrm{KL}}(q_\phi(z|v^{(i)}, I^{(i)})||p(z|v^{(i)}, I^{(i)}))$$
$$+ \mathbb{E}_{q_\phi(z|v^{(i)}, I^{(i)})}[\log p_\theta(v^{(i)}|I^{(i)}, z)] \tag{9}$$

$$\geq -D_{\mathrm{KL}}(q_\phi(z|v^{(i)}, I^{(i)})||p_z(z)) + \mathbb{E}_{q_\phi(z|v^{(i)}, I^{(i)})}[\log p_\theta(v^{(i)}|I^{(i)}, z)] \tag{10}$$

$$\triangleq \mathcal{L}(\theta, \phi, v^{(i)}|I^{(i)}) \tag{11}$$

The first KL-divergence term in Eq. 11 has an analytical form (see Kingma and Welling [2014] for details). To make the second term tractable, we approximate the variational distribution, $q_\phi(z|x^{(i)}, I^{(i)})$, by its empirical distribution. We have

$$\mathcal{L}(\theta, \phi, v^{(i)}|I^{(i)}) \approx -D_{\mathrm{KL}}(q_\phi(z|v^{(i)}, I^{(i)})||p_z(z)) + \frac{1}{L}\sum_{l=1}^{L}\left[\log p_\theta(v^{(i)}|z^{(i,l)}, I^{(i)})\right], \tag{12}$$

where $z^{(i,l)}$ are samples from the variational distribution $q_\phi(z|v^{(i)}, I^{(i)})$. Eq. 12 is the variation lower bound that our network minimizes during training.

In practice, we simply generate one sample of $z^{(i,l)}$ at each iteration (thus $L = 1$) of stochastic gradient descent, but different samples are used for different iterations.

## A.2 Detailed Network Structure

This section describes the details of our network structure. During training, our motion encoder (Figure 3(a)) takes two adjacent frames in time as input, both at resolution $128 \times 128$. The network then applies six $5 \times 5$ convolutional and batch normalization layers (number of channels are $\{96, 96, 128, 128, 256, 256\}$) to the concatenated images, with some pooling layers in between. The output has a size of $256 \times 5 \times 5$. The kernel encoder then reshapes the output to a vector, and splits it into a 3,200-dimension mean vectors and a 3,200-dimension variance vector, from which the network samples the latent motion representation $z$.

Next, the kernel decoder (Figure 3(b)) sends the $3,200 = 128 \times 5 \times 5$ tensor into two additional convolutional layers, each with 128 channels and a kernel size of 5. They are then split into four sets, each with 32 kernels of size $5 \times 5$.

Our image encoder (Figure 3(c)) operates on four different scaled versions of the input image $I$ ($256 \times 256$, $128 \times 128$, $64 \times 64$, and $32 \times 32$). At each scale, there are four sets of $5 \times 5$ convolutional and batch normalization layers (number of channels are $\{64, 64, 64, 32\}$), two of which are followed by a $2 \times 2$ max pooling layer.

Therefore, the output size of the four channels are $32 \times 64 \times 64$, $32 \times 32 \times 32$, $32 \times 16 \times 16$, and $32 \times 8 \times 8$, respectively. This multi-scale convolutional network allows us to model both global and local structures in the image, which may have different motions.

The core of our network is a cross convolutional layer (Figure 3(d)) which, as discussed in Section 4.1, applies the kernels learned by the kernel decoder to the feature maps learned by the image encoder, respectively. The output size of the cross convolutional layer is identical to that of the image encoder.

Our motion decoder (Figure 3(e)) starts with an up-sampling layer at each scale, making the output of all scales of the cross convolutional layer have a resolution of $64 \times 64$. This is then followed by one $9 \times 9$ and two $1 \times 1$ convolutional and batch normalization layers, with $\{128, 128, 3\}$ channels. These final feature maps are then used to regress the output difference image (Eulerian motion map).