[Reviews · NeurIPS 2016]

Reviewer 1

Summary

Paper proposes a model for synthesizing likely future frames form a single input image. The difference from the existing literature is that the paper proposes a probabilistic framework that allows sampling potentially a variety of future frames (from the conditional distribution). The model encodes images as feature maps and differential motion information as convolutional kernels. The two are then combined using proposed cross-convolution layer to produce final predictions. The conditional distribution of future frames is modeled using a conditional variational auto encoder and the predictions are made for the intensity changes (Eulerian motions) that are then combined with the original image to produce final predictions (as opposed to predicting raw pixels of the future image). The approach is illustrated on synthetic and more realistic datasets with promising results.

Qualitative Assessment

Overall the paper addresses an interesting problem and the formulation is also intriguing. However, lack of more through ablation studies makes it unclear which parts of the model are truly important. The baselines are also rather weak as only a very simple kNN approach is tested as the baseline. A more through evaluation would make the paper considerably stronger. Generally I actually liked the paper for its ideas and addressed problem, but would like to read authors rebuttal before finalizing the decision. > References Generally, references are adequate and well discussed. Activity forecasting is also related to the proposed problem, though, typically deals with individual agents and their trajectories as opposed to an image as a whole. I would suggest citing and discussing some of those works. A good overview is given in: Social LSTM: Human Trajectory Prediction in Crowded Spaces A. Alahi, K. Goel, V. Ramanathan, A. Robicquet, L. Fei-Fei, S. Savarese CVPR, 2016 > Assumption - An intrinsic assumption of the work, it appears, that predictions are made under assumption of effectively static camera (this appears to be the reason why camera motion is factored out in the experiments as a pre-processing step). I do not view this as a negative, but do think that stating this assumption more clearly earlier on would be helpful for the reader. - It is unclear from either motivation or the description why latent variables, z, in Figure 2 (d) are really necessary. The direct motion prediction (Figure 2(b)) could also be done presumably probabilistically, where given an image I a conditional distribution over the difference image could be modeled. What is the presumptive benefit of introducing latent variables z? I would like authors to address this in the rebuttal. > Typos - Page 3, bottom line, “frame without almost no motion” -> “frame with almost no motion” > Experiments Overall the experiments are interesting and compelling. However, the only baseline is rather weak (10 nearest neighbors to the input image with random sampling among the 10). In the very least a non-parametric generative model should be considered as a baseline. In particular, consider taking features of the image f(I) and a difference image and forming a KDE (e.g., Gaussian kernel on every sample) from joint samples to model p(I,v). This KDE can then be trivially conditioned (gaussian mixture conditioned is still a gaussian mixture) to obtain p(v|I). This conditional KDE can then be sampled or evaluated. This is still simple, but my guess is that it will work better then NN baseline and is also more inline with the premise of the paper.

Confidence in this Review

2-Confident (read it all; understood it all reasonably well)


Reviewer 2

Summary

The paper presents a method for synthesizing new frames using as input a single frame. This is done using an encoder network for motion, an encoder network for the image, a cross convolution layer, and a decoder. The motion encoder further contains a motion prior and a probabilistic frame predictor. Experiments show positive qualitative results on synthetic and simple real data, along with positive numerical results from a user study.

Qualitative Assessment

The paper is very well written and easy to read. The experiments are convincing (I really like the synthetic one) and thorough. One aspect I wish the authors had commented on is: how does the method fail? how can it be improved? The real data seems to work ok, but sometimes the result is a bit blurry. IF the paper is accepted, I think it would make it much stronger and useful if they included a section on failure cases, and future work. Another aspect I'm curious about is the applications. The authors mention "visual analogy", but the results are a single image, so it's hard to judge how feasible this is. While the idea is interesting, I also think it would be very good to hear the authors' opinion on what possible applications the method could have.

Confidence in this Review

2-Confident (read it all; understood it all reasonably well)


Reviewer 3

Summary

This paper proposes a method for generating a 'future' frame from a single one, using a generative probabilistic model trained on pairs of adjacent video frames. They model the conditional distribution of the Eulerian motion map (difference between frames) given the input image. They propose a two stage network, one that allows to identify the input frame and another one that encodes video motion. Finally, they test their results on three different data sets and quantitatively using Mechanical Turk, where people evaluate if the motion is 'real' or not, giving good results.

Qualitative Assessment

The paper is well written and interesting but it has a few overclaims in the abstract and introduction. For instance, the authors claim that they can generate a set of frames that follow a single image, to create a video. But in the 3.1 Problem formulation, they state that they predict a single frame. The problem is very different, since for more than one frame you need to take care of possible inestabilities that appear as time evolves. Moreover, they claim to test their method with real videos, but the 'real videos' are in fact over simplified, pre-segmentated. I suppose a challenge in this case is how to synthesize complex background with or without motion, which may be out of the scope of the paper. Thus I suggest, either claim that it works on simple videos, or show real scenarios. I also think that showing the cases where it fails would help focus future research, thus making this paper more appealing and understandable. There is an interesting idea in this paper: motion is a characteristic of a certain object when it is in a certain position. Therefore, a generative model needs to first be able to distinguish the object and its position in the initial frame. This makes it different from other methods based on lstm such as: Srivastava et al. 'Unsupervised Learning of Video Representations using LSTMs.' which model a certain motion. But anyway, the authors should discuss why this is a better or more interesting approach than LSTMs, which have been used previously. I have a technical question, is the method robust to translation? how does the method perform if the object is on the boundary of the image (thus motion leads outside the image). What are in general the limits of this next-frame generation? Another question is why use Mechanical Turk and not automatic action recognition on the generated frames. Even though the paper has a few changes to be made, I think it is interesting and thus suggest 'poster level'.

Confidence in this Review

2-Confident (read it all; understood it all reasonably well)


Reviewer 4

Summary

This paper address the problem of predicting future frames based on a single image. Rather than directly predicting the frames in a deterministic manner, the paper proposes to model the conditional probability of future frames given the input image. More specifically, it models the conditional probability of Eulerian motion given the input image. In practice, a motion encoder module generates sample specific kernels to model the motion patterns given the input image, the motion kernels convolves the image feature maps in the "cross convolution layer" to detect motions patterns on the images. The motion patterns are then decoded into Eulerian motion and thus get the final result.

Qualitative Assessment

Strengths: 1. The paper investigates in an interesting problem of modeling future frames given a single image. The problem is essentially modeled as a conditional distribution of local motion patterns given the input image and motion priors. Thus the proposed framework is able to generate multiple results given the same input to address the ambiguity problem. 2. The probability model of the problem is intuitive and theoretically sound. 3. The organization of the paper is easy to follow and the descriptions are concise and clear. 4. The network generates sample-specific convolution kernels to detect motion patterns, which is an interest direction. Doubts: 1. My biggest doubt about this paper is the motivation of the problem. The problem is essentially modeling conditional distribution of motion given the motion priors and appearance patterns. Compare to conventional frame predicting methods based on videos, the proposed method relies heavily on appearance while video-based methods relies more on motion. This raise question on the generation capability of the work on natural videos. The paper seems to only be evaluated synthetic videos (shapes) and well-aligned videos (sprites and workout), and in my opinion, should be at least applied on more general and challenge videos (such as UCF 101, Sports 1M, etc). 2. In terms of predicting future frames, long-term capability is usually an important aspect to evaluate the robustness and stability of the systems. However, long-term accuracy of the algorithms is less studied in this paper, only 1 frame is predicted. 3. The baseline methods are somewhat easy. There lacks a objective evaluation metric for non-synthetic datasets (sprites and workout). Trivia: Line 295: Figure 1 -> Table 1

Confidence in this Review

2-Confident (read it all; understood it all reasonably well)


Reviewer 5

Summary

A network model that uses a correlation layer (has no trainable parameters) to correlate image shapes with motions and, thus, predict future frames from a single frame.

Qualitative Assessment

Majors: p. 5 Why do you use 4 scales scaled at powers of 2? Do you have any theoretical or practical justification? E.g. show results with only 2 scales. Also the description of the overall architecture with four scales lacks details: How / when do you combine the four scales? Why do you up-scale to a resolution of 64 x 64? The input is 128 x 128. Would it not be natural to have a 128 x 128 predicted image size instead of 64 x 64? p. 5 None of the details for the training procedure were given. What is the protocol for training? Learning rate? Momentum term? Batch size? Number of training steps? What platform did you use? E.g. Tensorflow, theano, torch, …? “to the concatenated images, with some pooling layers”. Please be specific – “some pooling layers” is not. Also, your description of the network architecture is incomplete. What happens with the borders when you use convolutional layers? What is concatenated and where? Where do you have the two max-poolings exactly? Any non-linearities? p. 6 It is unclear from the description how the gradient travels in backward direction to both inputs? Some equations describing your correlation layer would be nice (if you need to cut, cut the lengthy introduction and the toy example). p. 7 For the video game sprites you have 140 pairs for testing. For the mechanical turk experiment you have 200 images. Where these different data sets? How many individuals did you have for labeling each of the 200 images? What did you do when labelers gave different answers for the test stimuli? E.g. labeler 1 and 2 felt the animation is artificial while labelers 3-10 felt the animation is natural. General: What about making predictions like the pose of cars as shown in Reed et al. 2015 or pose of faces? Minors: p. 2 How is Eulerian motion defined? I heard about optic flow, visual image motion, etc. (see BKP Horn's “Robot Vision” for a definition of these terms). p. 7 “An ideal algorithm should be able to achieve a success rate of 50%.” Why? Should it not be a 100%. p. 8 L 295 Table 1 instead of Figure 1. p. 9 L299 it automatically learn*ed* minimal bits.

Confidence in this Review

3-Expert (read the paper in detail, know the area, quite certain of my opinion)


Reviewer 6

Summary

This paper evaluates an approach to learning transformations from synthetic and real-world video such that possible transformations can be predicted from a single image outside of the training set (analogy making). A cross convolutional network is used to model transformations as intensity changes which aids in analogy making and and keeps the representation low dimensional. The results show that the network is able to grasp and produce motion from both synthetic and real-world video, containing both uniform/simple and non-uniform/complex regional motion. The results show high quantitative accuracy on the synthetic data at predicting motion by learning the underlying correlations between shapes and their motion. The results also show high qualitative accuracy on real-world data at predicting motion by learning underlying correlations between human-like features and their motion, based on crowdsourced polls comparing real transformations to the generated transformations.

Qualitative Assessment

The paper is well-written and the ideas come out very clearly. The tests are convincing with both real-world and animated humanoid motion. A test on non-humanoid motion would show more diversity, such as depth translation/rotations for more rigid bodies. Though the humanoid tests are probably more difficult, given that different regions will have different motions, it would be nice to see some more diversity in type of motion you look at. This would make the work much more impactful and useful for future researchers to build upon. Though it is likely to be future work, it would be nice to see the effectiveness of this algorithm on less clean real-world data, such as frames where the objects are not as isolated and there is more clutter. Also, how parameter-dependent is this? A discussion on the increase and decrease in accuracy relative to the network parameters would be informative.

Confidence in this Review

2-Confident (read it all; understood it all reasonably well)